# RefScale: Multi-temporal Assisted Image Rescaling in Repetitive Observation Scenarios

## ABSTRACT

With the continuous development of imaging technology and the gradual expansion of the amount of image data, how to achieve high compression efficiency of high-resolution images is a challenge problem for storage and transmission. Image rescaling aims to reduce the original data amount through downscaling to facilitate data transmission and storage before encoding, and reconstruct the quality through upscaling after decoding, which is a key technology to assist in high-ratio image compression. However, existing rescaling approaches are more focused on reconstruction quality rather than image compressibility. In repetitive observation scenarios, multi-temporal images brought by periodic observations provide an opportunity to alleviate the conflict between reconstruction quality and compressibility, that is, the historical images as reference indicates what information can be dropped at downscaling to reduce the information content in downscaled image and provides the dropped information to improve the image restoration quality at upscaling. Based on this consideration, we propose a novel multi-temporal assisted reference-based image rescaling framework (RefScale). Specifically, a referencing network is proposed to calculate the similarity map to provide the referencing condition, which is then injected into the conditional invertible neural network to guide the information drop at the downscaling stage and information fusion at the upscaling stage. Additionally, a low-resolution guidance loss is proposed to further constrain the data amount of the downscaled LR image. Experiments conducted on both satellite imaging and autonomous driving show the superior performance of our approach over the state-of-the-art methods.

## CCS CONCEPTS

• **Computing methodologies** → **Computer vision**.

## KEYWORDS

Deep learning, Image rescaling, Multi-temporal fusion, Invertible neural networks, Image compression

## 1 INTRODUCTION

With the rapid advancements in imaging technology, high-resolution (HR) images and videos carry more visually pleasing details, delivering great benefits to human visual entertainment. However,

**Unpublished working draft. Not for distribution.**

Permission to make digital or hard copies of all or part of this work for personal or classroom use is granted without fee provided that copies are not made or distributed for profit or commercial advantage and that copies bear this notice and the full citation on the first page. Copyrights for components of this work owned by others than the author(s) must be honored. Abstracting with credit is permitted. To copy otherwise, or republish, to post on servers or to redistribute to lists, requires prior specific permission and/or a fee. Request permissions from permissions@acm.org.

*ACM MM, 2024, Melbourne, Australia*

© 2024 Copyright held by the owner/author(s). Publication rights licensed to ACM.

ACM ISBN 978-x-xxxx-xxxx-x/YY/MM

https://doi.org/10.1145/nnnnnnn.nnnnnnn

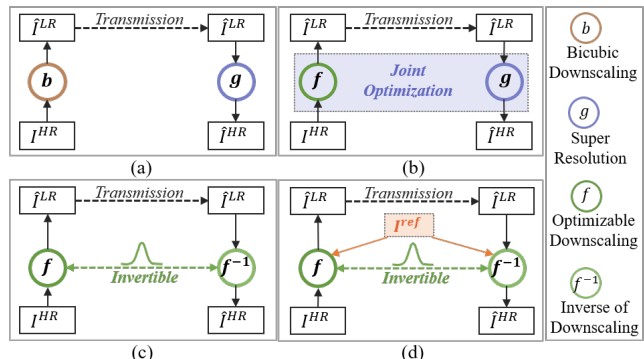

**Figure 1: Schematic of image rescaling frameworks. (a) fixed downscaling & SR methods, (b) joint-optimized downscaling and upscaling methods, (c) INN-based rescaling methods, and (d) our reference-based rescaling method.**

the huge amount of data has resulted in significant challenges related to data storage and transmission. Therefore, a high effective compression method is urgently required to alleviate the conflict between limited bandwidth and high data volume, which is crucial for the storage, transmission, and management of image data, and even for technologies like Internet of Things and Cloud Computing.

In order to achieve high-ratio compression, the rescaling-based coding, which contains rescaling module and compression module to form a Downscaling-Compression-Upscaling pipeline, stands out from other methods due to its low cost and high efficiency. Specifically, image downscaling is quite indispensable for storing and transferring large-size images, as the reduction of spatial resolution naturally removes part of the spatial information redundancy and significantly release the data amount. Additionally, when the content needs to be viewed, upscaling techniques are used to restore image details to their original resolution or adapt them to screens with varying resolutions. However, high-frequency details inevitably get lost during downscaling, posing a great challenge for reverse image reconstruction, which are generally referred to as "ill-posed" problems [4, 21, 36, 45].

To tackle the inverse problem of downscaling and upscaling, image rescaling has been studied mainly in three categories [42]. The first category is composed of methods that downscale images with fixed kernels, and upscale the LR images with image super-resolution (SR) techniques [3, 35, 49] (Figure 1(a)). Although advanced SR methods can restore some high-frequency textures, the crucial information lost during downscaling is hard to be fully recovered. With the insight that proper downscaling designs are influential in preserving beneficial information in LR images, the second category attempts to preserve the critical information for inverse restoration by optimizing both processes within a unified framework [9, 16, 33] (Figure 1(b)). They have achieved remarkable

visual reconstruction quality, but the authenticity of generated textures cannot be guaranteed. To fulfill the requirement of faithful image recovery, invertible neural network (INN) [2, 10, 32] based methods were proposed to build downscaling and upscaling into an invertible process [17, 43] (Figure 1(c)). Up till now, these rescaling methods have greatly boosted the quality of reconstructed HR images, but little attention has been paid to the data amount of their LR counterparts, which may be overly informative for storage and transmission.

This research aims to address the two challenges simultaneously: minimizing the amount of information conveyed in the downscaled images while maximizing the quality of the upscaled images. Inspired by the recent reference-based SR methods [24, 44, 50, 52], we propose leveraging multi-temporal images as historical reference to achieve this goal. Specifically, reference images could guide the removal of redundant information that is similar to them in the downscaling process and provide the dropped information from the reference in the upscaling process to compensate for the loss of details. This strategy can be deployed in repetitive observed scenarios, such as satellite imaging with repetitive observations at specific locations or autonomous driving with repeated excursions along predetermined paths, for reducing their data storage and transmission costs [1, 24, 41, 44, 52].

In this paper, we propose a multi-temporal assisted reference-based framework for image rescaling (RefScale) (Figure 1(d)) to alleviate the conflict between low information content and high visual quality by exploiting the relevance between the HR image and its historical reference. On one hand, during downscaling, we expect to drop as much mutual information that is comparable to reference images as possible, while preserving individual information that cannot be recovered from the reference. Additionally, the mutual information supplied by the reference can provide guidance for HR image reconstruction. To achieve this, we build a referencing network (Ref-Net) that collects the relevance between the HR and reference image and propose a novel conditional INN that embeds the mutual information into a latent variable that follows a specified distribution while retaining the individual information in the LR image. The HR image is then reconstructed based on the LR input, the condition, and random samples of latent variables. We employ a bottleneck structure and a quantization module to generate the similarity map with low bandwidth expense for recording the relevance. A similarity-based LR guidance loss is also designed to guide the redundancy elimination during training.

In summary, the main contributions of this paper are:

1) We introduce multi-temporal images as reference in the repetitive observation scenarios for image rescaling to achieve a joint optimization of compression-friendly downscaling and high-quality upscaling as dropping the mutual information between the reference and the current images helps to reduce information content in the downscaled image whilst the reference will bring back the dropped information for image reconstruction.

2) A novel rescaling framework called RefScale is proposed to integrate the reference in an invertible rescaling network, through the designs of referencing condition generation to indicate the similarity between the reference and the current image, and the condition-based downscaling and upscaling network to conditionally drop and recover the mutual information in the rescaling process.

3) Extensive experimental results demonstrate that our approach performs state-of-the-art results on both remote sensing and driving scenarios for both high reconstruction quality and low information content in LR images.

## 2 RELATED WORK

### 2.1 Image Rescaling

Image rescaling refers to the resizing of a digital image, including image downscaling and upscaling. Previous work typically treated the two processes separately. The widely used downscaling approaches employ high-frequency filters, such as bilinear and bicubic. Later methods [11, 23, 37] attempted to preserve more structure and details in downscaled images for higher visual quality. Kopf *et al.*[11] proposed a content-adaptive method to optmize the shape and locations of downsampling kernels. Oeztireli *et al.*[28] optimized downscaled images with structural similarity index (SSIM). Weber *et al.*[37] preserved visually significant details by emphasizing the distinctive pixels, while Liu *et al.*[23] introduced a gradient-ratio prior to preserving salient edges. On the other hand, the upscaling process is typically achieved by image SR techniques. Since Dong *et al.* [3] proposed the convolutional neural networks (CNNs)-based single image SR (SISR), various effective SR models [5, 14, 35] have been proposed for deeper models and higher accuracy, some of which developed more powerful modules, *e.g.*, residual connections [5, 14], dense connections [6, 35] for deeper models and higher accuracy. Suffering from blurry results, some methods [27, 46] employed generative adversarial networks (GANs) to produce visually more perceptible results. More recently, normalizing flow [25, 48] and diffusion models [29] have been introduced to SISR and achieved superior performance.

Recognizing the potential relevance between image downscaling and upscaling, recent studies [9, 16, 33] developed rescaling models that are jointly optimized for both processes and achieved more vivid reconstruction. Kim *et al.* [9] first proposed an autoencoder-based method, where the encoder and decoder simulate the downscaling and upscaling procedures, respectively. Li *et al.* [16] aimed to learn a compact-resolution image that is both visually pleasing and informative compared to HR images. Sun *et al.* [33] learned a content-adaptive downscaling kernel to maintain the structure of the HR input. More recently, INN-based methods [17, 43] were proposed to explicitly model the lost information during downscaling, resulting in remarkable reconstruction quality. IRN [43] was proposed to model the entire rescaling procedure using a bijective INN. HCFlow [17] further modeled the high-frequency component conditioned on the generated low-frequency component. Though these methods achieved outstanding reconstruction quality, they also introduced more information to LR images, which significantly increases the burden on data storage and transmission.

### 2.2 Reference-based Image Processing

The emergence of deep learning has led to significant advancements in computer vision tasks. However, there is still a lack of prior information learned from large-scale training data for restoration-related tasks such as image SR and image inpainting [19, 22, 34]. To break this dilemma, some methods have introduced reference images to provide auxiliary information. Reference-based image SR (RefSR)

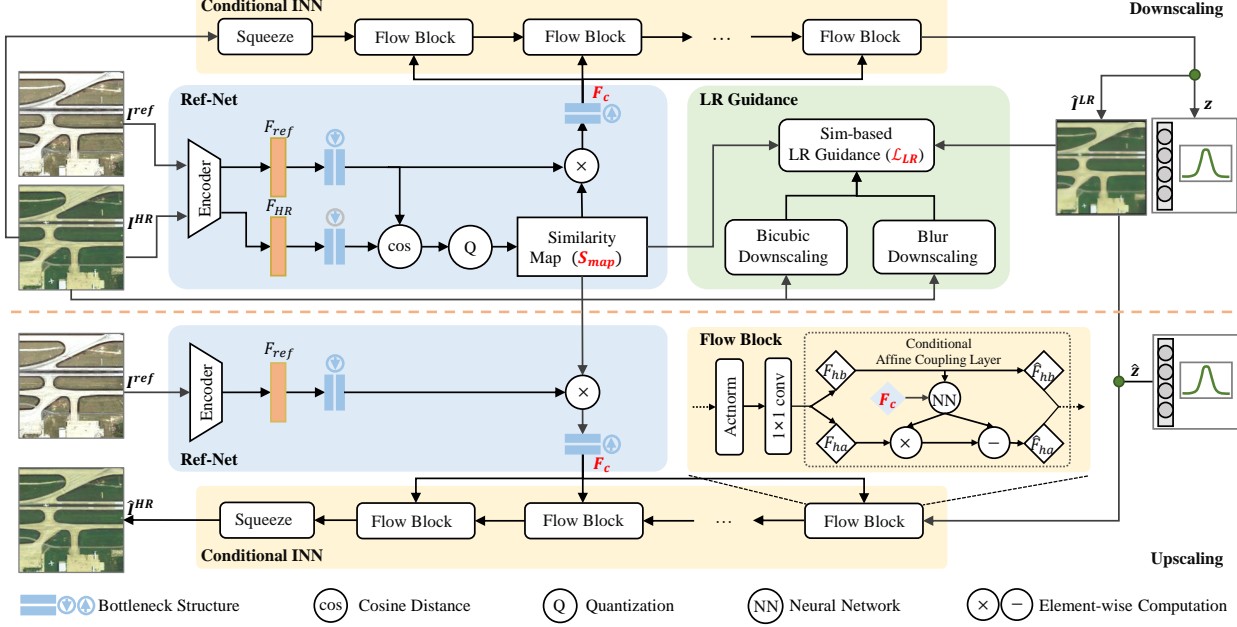

**Figure 2: Overview of our reference-based rescaling framework. At the downscaling stage, Ref-Net extracts the correlation between $I^{ref}$ and $I^{HR}$ as condition $F_c$, based on which $I^{HR}$ is downscaled to a less informative $\hat{I}^{LR}$ by conditional INN. At the upscaling stage, we store $S_{map}$ to extract the same $F_c$ as when downscaling and the conditional INN reconstructs the $\hat{I}^{HR}$ from $\hat{I}^{LR}$, $F_c$ and random sampled $\hat{z}$. Furthermore, a similarity-based LR guidance loss leads to an adaptively smooth $\hat{I}^{LR}$.**

methods [8, 24, 31, 44, 50, 52] transfer high-frequency details from the reference image to the super-resolved image and exhibit promising results over SISR. Image inpainting methods [7, 15, 51, 53] take reference images as guidance for realistic texture and structure inference, effectively alleviating artifacts and unreasonable contents caused by large holes. The reference also frees users from the laborious interaction process for image generation [13, 38–40]. While reference-based image processing algorithms call for more specific application scenarios, they have shown to be vastly superior to approaches with no reference.

## 3 PROPOSED METHOD

### 3.1 Problem Formulation

Let $I^{HR} \in \mathbb{R}^{3 \times W \times H}$ be the HR image, and $I^{LR} \in \mathbb{R}^{3 \times \frac{W}{s} \times \frac{H}{s}}$ be the corresponding LR image, where $W$ and $H$ denote the width and height, respectively, and $s$ is the downscaling factor. Conventionally, $I^{LR}$ is obtained using a predefined downscaling kernel, such as bicubic. Recent rescaling studies aim to jointly learn and optimize the downscaling operation $f(\cdot)$ and the upscaling operation $g(\cdot)$ with respect to the objective of superior quality of the downscaled image $\hat{I}^{LR}$ and the reconstructed HR image $\hat{I}^{HR}$, given by:

$$\mathcal{L}_{img} = \|I^{HR} - \hat{I}^{HR}\|^2 + \lambda \|I^{LR} - \hat{I}^{LR}\|^2, \quad (1)$$

where $\lambda$ is a weight that balances the two terms.

In addition to the image quality constraints, our objective is to reduce the information content (IC) contained in $\hat{I}^{LR}$ to save on storage and transmission costs. To this end, we introduce a term in the objective function that constrains the information entropy of $\hat{I}^{LR}$ and encourages dropping more details during downscaling,

expressed as:

$$\mathcal{L}_{img+ic} = \mathcal{L}_{img} + \lambda_e E(\hat{I}^{LR}), \quad (2)$$

where $E(\cdot)$ measures the IC in $\hat{I}^{LR}$ and $\lambda_e$ is the weight. The newly introduced item encourages dropping more details during downscaling, which hinders the reconstruction process. Thus, the key to resolving the problem is to fully use the information contained in the reference image.

### 3.2 Overall Framework

In this paper, we introduce a reference image $I^{ref} \in \mathbb{R}^{C \times W \times H}$ and exploit the correlations between $I^{HR}$ and $I^{ref}$ for the image rescaling task. The proposed reference-based rescaling framework (RefScale) is depicted in Figure 2, which consists of two sub-networks: a referencing network (Ref-Net) to generate a referencing condition from the HR image and the reference image, and then a conditional invertible neural network (INN) adopted the reference condition to perform image downscaling and upscaling respectively. A similarity-guided LR guidance loss is proposed to further constrain the IC of the downscaled image.

In particular, at the downscaling stage, the Ref-Net calculates a similarity map $S_{map}$ from $I^{HR}$ and $I^{ref}$, which is subsequently used to generate the referencing condition feature $F_c$. $S_{map}$ is also used to guide the formation of the LR guidance loss $\mathcal{L}_{LR}$, under the consideration that the information can be more eliminated at similar regions while more information should be kept at dissimilar regions. Then, with the guidance from $F_c$ and $\mathcal{L}_{LR}$, the conditional INN processes $I^{HR}$ to generate a less informative LR image $\hat{I}^{LR}$ and embed the mutual information of $I^{HR}$ and $I^{ref}$ into a latent variable

$z$, which follows a specific distribution. At the upscaling stage, Ref-Net takes $I^{ref}$ and $S_{map}$ as inputs to generate the condition $F_c$, whilst the conditional INN samples the latent variable $\hat{z}$ so as to inversely reconstruct the HR image from the LR image $\hat{I}^{LR}$ by the referencing condition $F_c$. As $I^{HR}$ is inaccessible, the similarity map $S_{map}$ needs to be transmitted with $\hat{I}^{LR}$ from the downscaling end to the upscaling end, thus efforts have also been made to reduce the size of $S_{map}$.

In the following, we illustrate the key modules of the proposed RefScale, including: 1) the referencing network generating the referencing condition, 2) the LR guidance using the referencing similarity to constrain the IC of LR, and 3) the conditional INN to transfer between HR and LR images under the referencing condition and the LR guidance.

## 3.3 Referencing Network

The Ref-Net captures the correlations between $I^{HR}$ and $I^{ref}$ and generates a similarity map $S_{map}$ for LR guidance and a condition feature $F_c$ for conditional INN.

*Extraction of Similarity Map $S_{map}$ and Condition Feature $F_c$.* As depicted in Figure 2, $I^{HR}$ and $I^{ref}$ are processed by the same encoder, and the output features $F_{HR}$ and $F_{ref}$ are used to calculate $S_{map}$ by the pixel-wise cosine distance operation:

$$S_{map}^{i,j} = \left\langle \frac{F_{HR}^{i,j}}{\|F_{HR}^{i,j}\|}, \frac{F_{ref}^{i,j}}{\|F_{ref}^{i,j}\|} \right\rangle, \tag{3}$$

where $(i, j)$ is spatial position index. The referencing condition feature $F_c$ is obtained by combining $F_{ref}$ with $S_{map}$:

$$F_c = F_{ref} \odot S_{map}, \tag{4}$$

where $\odot$ denotes element-wise multiplication.

*Shrinkage of Similarity Map $S_{map}$.* It should be noted that $I^{HR}$ is not available at the upscaling end, which means that $S_{map}$ needs to be preserved and transmitted to the downscaling end to generate the same condition with the accessible reference image $I^{ref}$. To achieve this, we employ a bottleneck structure and a quantization module to reduce the size of $S_{map}$. First, $F_{HR}$ and $F_{ref}$ are processed by downsampling layers to obtain a lower resolution for calculating $S_{map}$. Second, during the calculation of $F_c$, $S_{map}$ is processed by upsampling layers to match the spatial size to $F_c$. Then, the quantization module is used to convert $S_{map}$ from floating point to integer to further reduce the data size. As quantification is not differentiable, we introduce uniform noise of $U(-0.5, 0.5)$ during training to simulate this operation and quantify the similarity map when verifying our model.

## 3.4 Low-Resolution Guidance

The mutual information extracted by Ref-Net tells the conditional INN that what can be discarded and recovered. In this section, we present a LR guidance to tell the conditional INN how to discard them, targeting reducing the IC of the LR image in similar regions with the reference while keeping details in dissimilar regions.

*Similarity-based LR Image Composition.* Following the rescaling studies that utilize a bicubic-based downscaled image to guide LR image generation, we keep the bicubic as a detail-informative guide for dissimilar regions since existing methods can already reconstruct bicubic images well. Additionally, considering similar information between image pairs primarily pertains to structural features, we propose a new guide for similar regions, namely a Gaussian blurred downscaled image, which is less informative but still visually recognizable. It is processed by a downscaling and Gaussian blurring pipeline, which results in a significant loss of details and reduces the amount of information content. This new kind of LR guidance is adopted for constraining the downscaled image by a similarity-based LR guidance loss, which is formulated as follows:

$$\mathcal{L}_{LR} = \left\| (I_{bic}^{LR} - \hat{I}^{LR}) \odot (1 - S_{map}) \right\|^2 + \left\| (I_{blur}^{LR} - \hat{I}^{LR}) \odot S_{map} \right\|^2, \tag{5}$$

where $I_{bic}^{LR}$ and $I_{blur}^{LR}$ are bicubic-based and blur-based downscaled images, respectively. This loss is to drive $\hat{I}^{LR}$ to discard more information where the HR image is similar to the reference but keeps irrecoverable details in $\hat{I}^{LR}$.

## 3.5 Conditional Invertible Neural Network

To accomplish this goal of integrating the referencing condition from the reference, we propose a conditional invertible transformation $f(\cdot)$: $I^{HR} \overset{c}{\leftrightarrow} [I^{LR}, z]$, where $c$ is the mutual information between the HR and reference images, and $z$ is a latent variable. The probability of HR images conditional on the mutual information is expressed as:

$$p(I^{HR}|c) = p(I^{LR}, z|c) \left| \frac{\partial(I^{LR}, z)}{\partial I^{HR}} \right|$$
$$= p(I^{LR}|c) p(z|c) \left| \frac{\partial f(I^{HR})}{\partial I^{HR}} \right|. \tag{6}$$

Here, we assume that $I^{LR}$ is independent with $z$ as in IRN [43] and expect to exclude the information of $z$ from the conditional probability of $I^{LR}$ given $c$ to exclude the information of $z$. To this end, we approximate it as a multivariate Gaussian distribution with a mean of $\tilde{I}^{LR}$, which is a LR image that excludes mutual information. Furthermore, we formulate $p(z|c)$ as a standard multivariate Gaussian distribution, then the model can be defined as:

$$p(I^{HR}|c) = \mathcal{N}(I^{LR}; \tilde{I}^{LR}, \Sigma) \mathcal{N}(z; \mathbf{0}, \mathbf{1}) \left| \frac{\partial(I^{LR}, z)}{\partial I^{HR}} \right|, \tag{7}$$

where $\Sigma$ is a diagonal covariance matrix with all diagonal elements close to zero. In practice, we employ the downscaled LR image as $\tilde{I}^{LR}$ and the referencing condition $F_c$ as $c$.

*Architecture of Conditional INN.* We design a conditional INN with a newly introduced conditional affine coupling transform [10], as illustrated in Figure 2. The proposed conditional INN consists of a squeeze block and a stack of conditional flow blocks (CFB). The squeeze block exchanges spatial size for channel numbers to align the spatial dimension with the LR image. The CFB, which includes an Actnorm layer, an invertible convolution layer, and a conditional

**Table 1: Quantitative results of different downscaling and upscaling methods for image reconstruction on RS and ACDC-snow datasets. † denotes the method can only implement 4× upscaling. We report the mean result of our method over 5 draws of $z$.**

| Category | Method | RS Dataset | | | | | | | | ACDC-snow Dataset | | | | | | | |
|---|---|---|---|---|---|---|---|---|---|---|---|---|---|---|---|---|---|
| | | 2× | | | | 4× | | | | 2× | | | | 4× | | | |
| | | PSNR↑ | SSIM↑ | SI↓ | bpp↓ | PSNR↑ | SSIM↑ | SI↓ | bpp↓ | PSNR↑ | SSIM↑ | SI↓ | bpp↓ | PSNR↑ | SSIM↑ | SI↓ | bpp↓ |
| SISR | Bicubic & Bicubic | 28.82 | 0.878 | 0.454 | 2.55 | 23.21 | 0.658 | 0.561 | 3.12 | 32.26 | 0.943 | 0.261 | 1.67 | 24.93 | 0.769 | 0.319 | 2.03 |
| | Bicubic & EDSR | 30.90 | 0.903 | 0.454 | 2.55 | 25.04 | 0.727 | 0.561 | 3.12 | 34.88 | 0.959 | 0.261 | 1.67 | 27.06 | 0.836 | 0.319 | 2.03 |
| | Bicubic & RCAN | 32.48 | 0.908 | 0.454 | 2.55 | 26.41 | 0.719 | 0.561 | 3.12 | 36.04 | 0.958 | 0.261 | 1.67 | 28.26 | 0.833 | 0.319 | 2.03 |
| | Bicubic & NLSN | 31.07 | 0.906 | 0.454 | 2.55 | 25.17 | 0.722 | 0.561 | 3.12 | 34.80 | 0.959 | 0.261 | 1.67 | 27.15 | 0.838 | 0.319 | 2.03 |
| | Bicubic & LDL | 31.77 | 0.872 | 0.454 | 2.55 | 26.24 | 0.694 | 0.561 | 3.12 | 40.56 | 0.955 | 0.261 | 1.67 | 30.23 | 0.800 | 0.319 | 2.03 |
| | Bicubic & LTE | 31.18 | 0.908 | 0.454 | 2.55 | 26.54 | 0.719 | 0.561 | 3.12 | 38.95 | 0.949 | 0.261 | 1.67 | 31.90 | 0.799 | 0.319 | 2.03 |
| RefSR | Bicubic & TTSR† | - | - | - | - | 26.74 | 0.737 | 0.561 | 3.12 | - | - | - | - | 29.36 | 0.847 | 0.319 | 2.03 |
| | Bicubic & $C^2$ Matching† | - | - | - | - | 27.62 | 0.773 | 0.561 | 3.12 | - | - | - | - | 29.81 | 0.859 | 0.319 | 2.03 |
| Joint-optimizing | TAD & TAU | 31.62 | 0.897 | 0.442 | 2.46 | 26.13 | 0.709 | 0.557 | 3.10 | 36.27 | 0.961 | 0.247 | 1.59 | 28.53 | 0.847 | 0.308 | 1.98 |
| INN-based | IRN | 38.22 | 0.983 | 0.487 | 2.59 | 30.66 | 0.876 | 0.569 | 3.13 | 40.59 | 0.988 | 0.228 | 1.44 | 33.79 | 0.944 | 0.296 | 1.78 |
| | HCFlow† | - | - | - | - | 30.92 | 0.881 | 0.573 | 3.14 | - | - | - | - | 33.91 | 0.946 | 0.301 | 1.81 |
| RefScale | **Ours** | **43.63** | **0.994** | **0.385** | **2.03** | **31.31** | **0.897** | **0.451** | **2.37** | **47.25** | **0.995** | **0.166** | **1.14** | **34.42** | **0.957** | **0.184** | **1.26** |

affine coupling layer, further removes mutual features and separates $\hat{I}^{LR}$ based on the referencing condition $F_c$. As all of these modules are invertible, thereby guaranteeing the invertibility of the entire INN. The entire downscaling and reverse reconstruction processes can be denoted as:

$$[\hat{I}^{LR}, z] = f(I^{HR}|F_c), \quad (8)$$

$$\hat{I}^{HR} = f^{-1}(\hat{I}^{LR}, \hat{z}|F_c). \quad (9)$$

*Referencing Condition Injection in CFB.* $F_c$ is injected into the conditional affine coupling layer of CFB to contribute to the process of discarding IC. The data flow of conditional INN progressively drops the information that can be obtained from $I^{ref}$ based on $F_c$. Specifically, in the conditional affine coupling layer, the input hidden feature flow $F_h$ is split into two sub-features $F_{ha}$ and $F_{hb}$ along the channel axis, and then both undergo the affine transform to decide what information should be discarded under the constraint of $F_c$:

$$\hat{F}_{ha} = F_{ha} \odot \exp(\psi(F_{hb}, F_c)) - \phi(F_{hb}, F_c), \quad (10)$$

$$\hat{F}_{hb} = F_{hb}, \quad (11)$$

where $\psi(\cdot)$ and $\phi(\cdot)$ are learnable functions and $[\hat{F}_{ha}, \hat{F}_{hb}]$ are the output features. $F_{hb}$ and $F_c$ are concatenated and fused by $\psi(\cdot)$ and $\phi(\cdot)$ to generate the parameters of affine transform, which is then applied to $F_{ha}$ to discard the information that can be recovered using $F_c$ and $F_{hb}$.

### 3.6 Training Objectives

Theoretically, INN can be trained by minimizing the negative log-likelihood loss. However, this loss cannot provide strong supervision for image reconstruction. Following IRN [43], we constrain both the reconstruction quality of $\hat{I}^{HR}$ and the distribution of $z$. Moreover, we use the similarity-based LR guidance on $\hat{I}^{LR}$ to ensure that the information discarded during downscaling can be recovered from the reference. The objectives of HR reconstruction and distribution matching are described below.

*HR Reconstruction.* We employ the widely used $\mathcal{L}_1$ loss to measure the difference between the reconstructed HR image and the ground truth, as specified below:

$$\mathcal{L}_{HR} = \|I^{HR} - \hat{I}^{HR}\|. \quad (12)$$

*Distribution Matching.* The objective of distribution matching is to encourage the generated latent variable to confirm a specific distribution:

$$\mathcal{L}_{distr} = CE(p(z), f^z[q(I^{HR})]), \quad (13)$$

where $q(I^{HR})$ and $p(z)$ are the distributions of HR image $I^{HR}$ and latent variable $z$, respectively, $f^z(\cdot)$ is the partial transformation of mutual information and $CE(\cdot)$ is cross entropy function. We assume that $p(z)$ follows a Gaussian distribution, thus the distribution matching can be easily calculated by $\mathcal{L}_2$ regularization on the latent variable $z$.

*Total Loss.* We optimize our model by minimizing the total loss composed by HR reconstruction loss $\mathcal{L}_{HR}$, LR guidance loss $\mathcal{L}_{LR}$, and distribution matching loss $\mathcal{L}_{distr}$:

$$\mathcal{L}_{total} = \lambda_{HR}\mathcal{L}_{HR} + \lambda_{LR}\mathcal{L}_{LR} + \lambda_{distr}\mathcal{L}_{distr}, \quad (14)$$

where $\lambda_{HR}$, $\lambda_{LR}$ and $\lambda_{distr}$ are the associated weights.

## 4 EXPERIMENTS

### 4.1 Experimental Setup

*Datasets.* To validate the effectiveness of RefScale, we utilize a dataset from the remote sensing scenes with readily available reference images. In addition, we evaluated on a driving scenes with repeated driving along a predetermined rout to demonstrate the extensibility of the proposed method. 1) **Remote sensing (RS)** dataset with historical images from periodical revisiting. We collected these 8-bit RGB images from the SPOT-5 satellite from various cities and timestamps, and their original resolution ranges from 1878×1400 to 6264×3456. We collected images of different scenes, such as railway station, port, airport, suburb, mountain and city center. Besides, the reference images are selected from different seasons or different years to guarantee the reliability of verification. To construct the training set, we crop them into non-overlapping patches of size 512×512 size, resulting in 6,690 image pairs of HR image and its reference. For the test set, we use 58 uncropped images with five historical images taken at different times serving as the reference

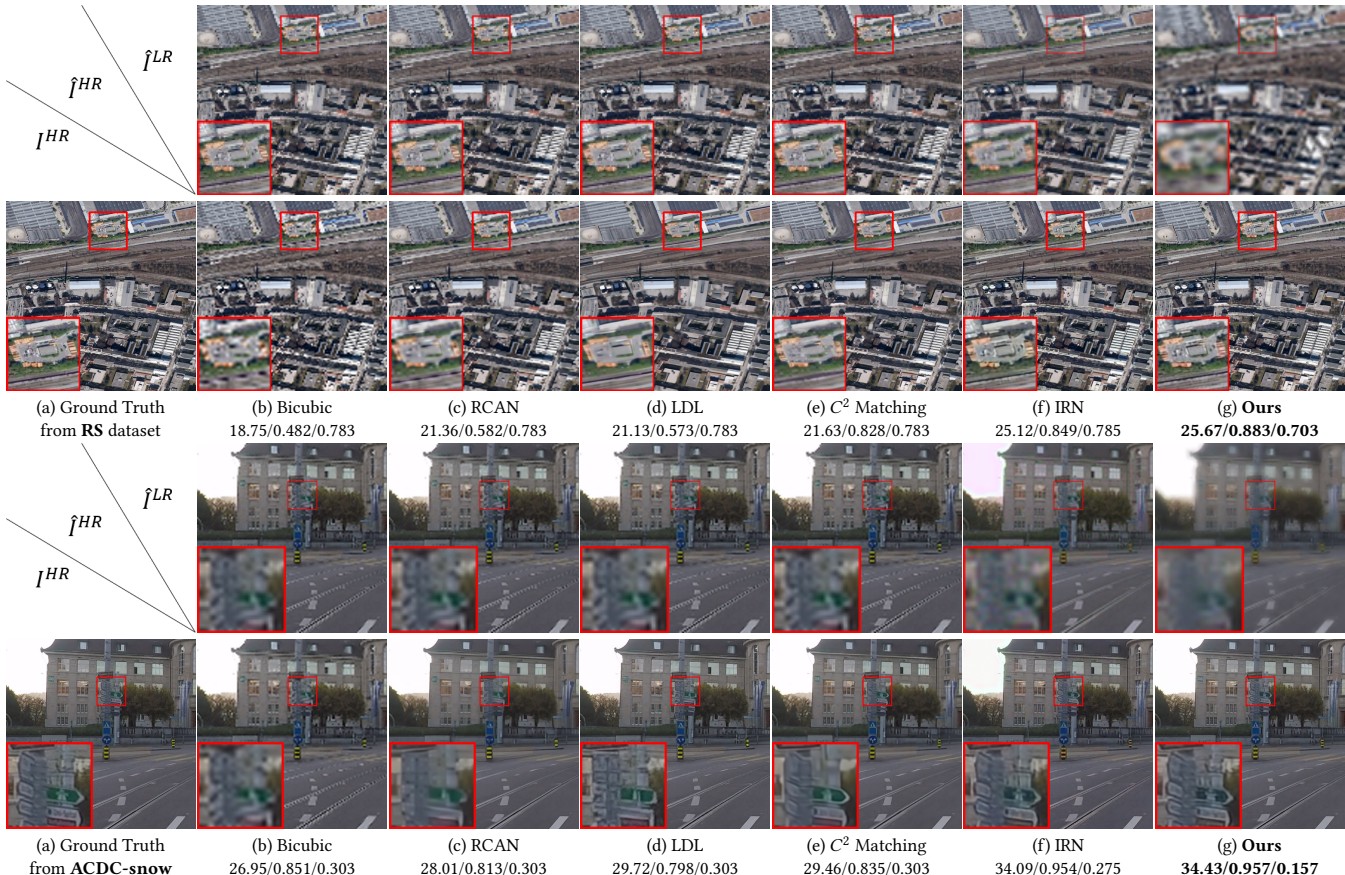

Figure 3: Qualitative results of RS and ACDC-snow images. The first row is downscaled LR images and the second row is corresponding restored HR images. PSNR/SSIM/SI are tagged below each method. Our method can generate smooth LR images and recover more details.

images. 2) **ACDC-snow** dataset [30] with reference images from re-visiting routes at different weather conditions. It is a natural image dataset consisting of cityscape driving images captured from the same routes but with different conditions, *i.e.*, one image captured in sunny day and another in a snowy day. It contains 1,000 images and is split into train set, validation set and test set for roughly 4:1:5 proportion, and the resolution of each image is 1920×1080.

*Baseline Methods.* We utilized three categories of image resizing methods as our baselines for comparison: 1) General SR methods, including six SISR methods, *i.e.*, Bicubic, EDSR [20], RCAN [49], NLSN [26], LDL [18], LTE [12], and two RefSR methods, *i.e.*, TTSR [44] and $C^2$ Matching [8], with bicubic interpolation for downscaling, 2) jointly optimized rescaling methods, TAD&TAU [9], and 3) INN-based rescaling methods, including IRN [43] and HCFlow [17].

*Evaluation Metrics.* We employ PSNR and SSIM for the comparison of reconstruction quality. We also adopt spatial information (SI) [47] as a metric to measure the image complexity of LR images. SI has been proved to be strongly positively correlated with JPEG-based image complexity measures. Specifically, it can be calculated by:

$$SI = \sqrt{s_h^2 + s_v^2}, \tag{15}$$

where $s_h$ and $s_v$ denote data of Y channel filtered with horizontal and vertical Sobel kernels, respectively. Additionally, we compress LR images and similarity maps and use bit per pixel (bpp) as an auxiliary metric for SI. the image as a long-term reference does not need to be transmitted, namely not to be included in the calculation of bpp.

*Implementation Details.* We implement the proposed RefScale with Pytorch and optimize it using Adam with $\beta_1 = 0.9$, $\beta_2 = 0.999$, and a learning rate of $2 \times 10^{-4}$. During training, the HR images and corresponding reference images are cropped to $128 \times 128$ patches and augmented with the same random flips and rotations. The batch size is set to 8. The loss weights $\lambda_{HR}$, $\lambda_{LR}$, and $\lambda_{distr}$ are set to 1, 4, 1 for 2× rescaling, and 1, 16, 1 for 4× rescaling, respectively. We employ RRDB [35] as the backbone of Ref-Net and initialize it by pre-training on natural images.

## 4.2 Experimental Results

*Quantitative Results.* To ensure a fair comparison, we fine-tuned all the baseline methods on both **RS** and **ACDC-snow** datasets. As shown in Table 1, our method achieves superior performance on both datasets, with increments in PSNR and SSIM and decreases in

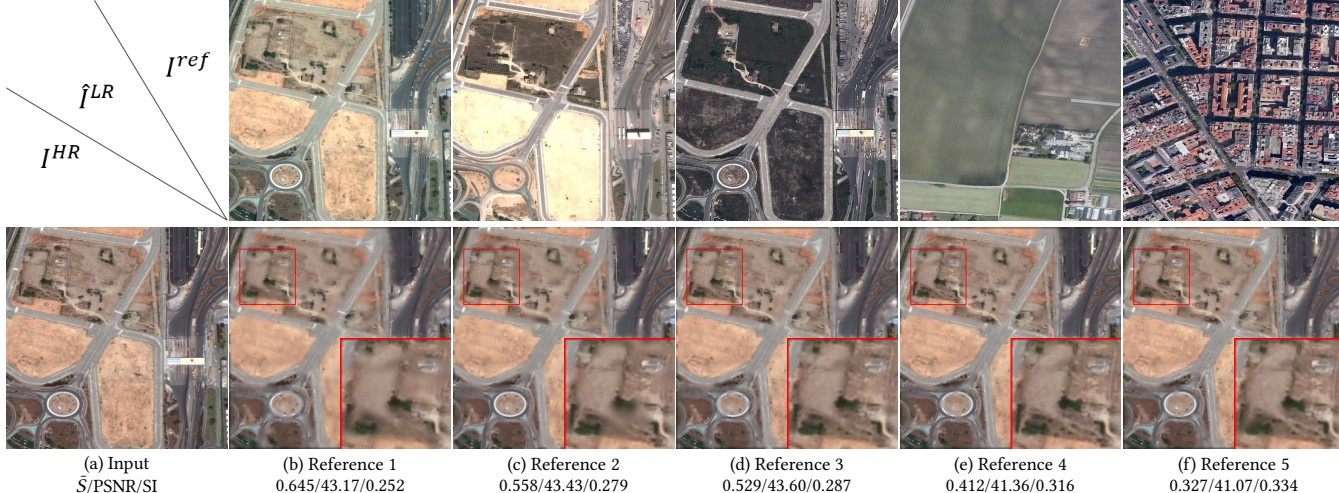

|   | (a) Input
$\bar{S}$/PSNR/SI | (b) Reference 1
0.645/43.17/0.252 | (c) Reference 2
0.558/43.43/0.279 | (d) Reference 3
0.529/43.60/0.287 | (e) Reference 4
0.412/41.36/0.316 | (f) Reference 5
0.327/41.07/0.334 |

**Figure 4: The effects of different reference images. The first row is the reference image with different similarity levels, whose similarity is measured by the average of similarity map as $\bar{S}$. The second row is the corresponding downscaled image. Our method tends to discard more details as the increasing of similarity.**

**Table 2: Ablation study the influence of condition premise (Cond.), quantification (Q), and resolution of similarity map structure (Res.) on referencing condition extraction.**

| Case | Cond. | Q | Res. | RS | | ACDC-snow | |
|---|---|---|---|---|---|---|---|
| | | | | PSNR | bpp$_{smap}$ | PSNR | bpp$_{smap}$ |
| 1 | - | - | - | 37.09 | - | 40.32 | - |
| 2 | Ref. | ✗ | $1 \times 1$ | 40.02 | - | 43.65 | - |
| 3 | Sim. | ✗ | $1 \times 1$ | 44.63 | - | 48.17 | - |
| 4 | Sim. | ✓ | $1 \times 1$ | 44.57 | 0.211 | 48.13 | 0.121 |
| 5 | Sim. | ✓ | $1/2 \times 1/2$ | 43.63 | 0.068 | 47.25 | 0.036 |
| 6 | Sim. | ✓ | $1/4 \times 1/4$ | 41.84 | 0.024 | 45.02 | 0.013 |

SI and bpp. Although the joint-optimizing and INN-based rescaling methods outperform the general SR methods in terms of image quality, they result in increased SI values on the RS dataset. Comparatively, our method can better recover image details and generate smoother LR images with a significant reduction in SI, which is beneficial for storage and transmission.

*Qualitative Results.* To illustrate the details of the downscaled and upscaled images, we visualize some samples from both datasets at a 4× scale. As shown in Figure 3, our method produces sharper and more realistic HR images, which contain more details and are more faithful to the ground truth than those reconstructed by baseline methods, since reference images provide rich auxiliary information. Moreover, our model leverages the reference image to guide the downsampling process and discard mutual information, which results in downscaled images that are smoother and contain fewer details than other methods.

## 4.3 Ablation Study

*Visualisation of Similarity Map.* To depict how Ref-Net captures the correlations, we present the similarity map in Figure 5. The map shows that our method can capture similar information from reference images that share a similar texture or structure with the HR images. The warm region in Figure 5 indicates the presence of

abundant similarity between the reference and HR images. It can be observed that proposed method effectively detects dissimilar regions, such as roads in the first row and a newly built airport runway in the second row. In this case, our method can preserve the individual information in LR images for better restoration.

*Analysis on LR Guidance.* To evaluate the effectiveness of the similarity-based LR guidance loss, we perform comparative experiments using different types of downscaled image guidance, generated by a weighted average of bicubic downsampled image and Gaussian blurred image, namely $\tilde{I}^{LR} = \omega \times I_{bic}^{LR} + (1-\omega) \times I_{blur}^{LR}$, ($\omega = 0, 0.25, 0.5, 0.75, 1$). The results are shown in Figure 6 and measured by PSNR and SI. The results from our similarity-based weighting scheme are on the upper left of the corresponding curves in both datasets, showing that our scheme can achieve better PSNR and lower SI simultaneously.

*Influence of Different References.* To validate the impact of reference differences, we employ three reference images of varying styles and two reference images of varying scenes, as illustrated in Figure 4. Reference 1-3 are images taken at the same location but on different dates, whereas Reference 4-5 are unrelated images. The degree of similarity between each reference image and the input image was measured using the average of the similarity matrix, denoted as $\bar{S}$. Reference 1 has the highest degree of similarity, resulting in the lowest SI, while References 4 and 5 have the highest SI since they are completely unrelated to the input image and thus the LR image needs to record all necessary information for reconstruction. Despite the diverse reference images, our method achieves stable reconstruction quality (b-d), demonstrating that it can adaptively discard information according to the given reference.

*Analysis on the Condition Components.* We explore the effects of referencing conditions with various settings, including condition components and the shrinkage of the similarity map. The evaluations are performed in terms of reconstruction quality and SI of the similarity map, as shown in Table 2. *Case 1* shows the result

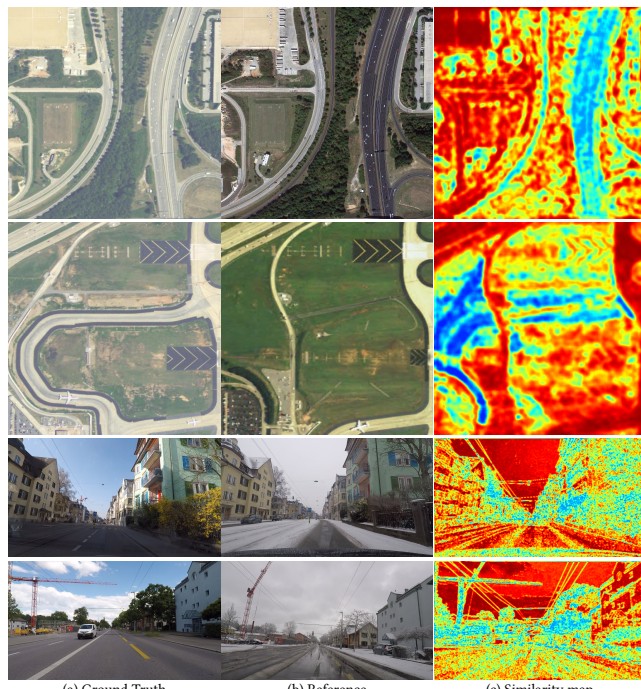

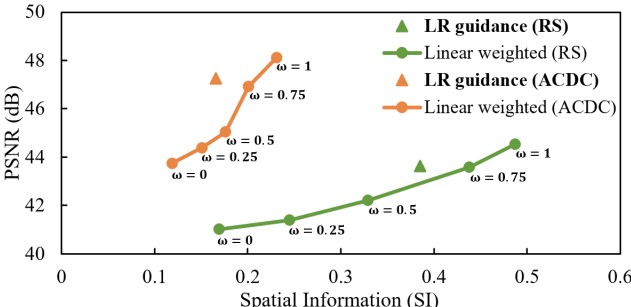

(a) Ground Truth        (b) Reference        (c) Similarity map

**Figure 5: Visualisation of similarity map. (a) and (b) are the input image and reference image. (c) is the visualization of a similarity map and the color from blue to red corresponds to the similarity value from low to high.**

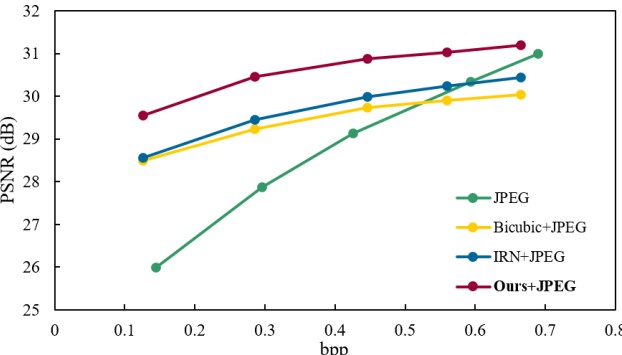

**Figure 7: Results of combination between rescaling and lossy image compression methods. We adjust the quality of JPEG to obtain different compression ratios. The scale of rescaling methods is set to 2×.**

the main factor, as refined correlations might be corrupted during pooling. *Case 5* results in acceptable quality degradation and IC, while the result of *Case 6* is similar to that of *Case 2*. Therefore, we take *Case 5* as the final reference setting.

## 4.4 Combination with Image Compression

We evaluate the combination of image rescaling and image compression methods on our RS dataset. We compare two rescaling method, bicubic interpolation and another INN-based method, IRN, and the scale of all rescaling methods is set to 2×. For a fair comparison, we adopt the JPEG algorithm as the combined lossy compression method since it requires no tuning. We adjust the quality of to control the similar compression ratio under different rescaling methods. The PSNR of the reconstructed image and bit rate are evalusted, and the R-D curves are shown in Fig. 7. As expected, the reconstruction quality of the rescaling-based compression methods outperforms that of the native compression method at low bit rates. IRN presents a slight advantage over bicubic interpolation, while the proposed Refscale shows a significant improvement over other methods, benefiting from the introduction of reference information.

## 5 CONCLUSION

In this paper, we propose RefScale, a reference-based image-rescaling framework for producing higher-quality reconstructed images with lower information content in downscaled images. RefScale involves a referencing network that exploits the correlations between the HR image and the reference to generate a similarity map and produce the referencing condition, which guides the generation of the LR image. We also propose a conditional INN that incorporates the mutual information into a latent variable, while retaining individual information in the LR image conditioned on the reference. Experiments demonstrate that RefScale outperforms the state-of-the-art methods in terms of both image reconstruction quality and data amount, as well as the lightweight property and high efficiency of our model. In addition, The proposed method is expected to assist high ratio compression to boost transmission efficiency, and show potential applications in scenarios where long-term observations are routinely conducted.

**Figure 6: Ablation study on LR guidance loss. The curves denote the results of linear weighted bicubic and blur guidance with the weight $\omega$ chosen uniformly between 0 and 1. Our result in a triangle is on the upper left of the curve, indicating that our loss can achieve higher PSNR and lower SI.**

without referencing condition, and the model degenerates into an INN-based rescaling model. In *Case 2*, the features of the reference image are used as conditions, and the performance boost is brought by introducing reference images. By utilizing the correlation information in *Case 3*, our model much outperforms the pure method in *Case 2*, but it lacks the quantization module, resulting in a high expense of storing and transferring the similarity map. *Cases 4* to *6* show the effects of quantization and bottleneck structure on the quality degradation and shrinkage of the similarity map. We find that quantization has a small effect on image reconstruction quality by comparing *Case 3* and *4*, while the bottleneck structure is

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
