# OpenReview forum: "RefScale: Multi-temporal Assisted Image Rescaling in Repetitive Observation Scenarios"
_acmmm.org/ACMMM/2024/Conference — MM2024 Poster_

### Official Review · Reviewer_P5mJ · 2024-05-18

**Rating:** 4
**Confidence:** 3

**Summary:**

The paper proposes a reference-based image-rescaling framework to minimize the amount of information conveyed in the downscaled
images while maximizing the quality of the upscaled images. A referencing network is proposed to exploit the correlations between the HR
image and the reference to generate a similarity map. Also, a conditional INN is proposed to incorporate the mutual information into a latent variable, while retaining individual information in the LR image conditioned on the reference.

**Strengths:**

+ The idea of using multi-temporal images as references to reduce the information conveyed in the downscaled images is interesting.
+ This paper is well-structured and is presented understandably with a logical flow between sections.
+ This paper presents comprehensive experiments.

**Limitations:**

- The papers mentioned in the related work section " Image Rescaling" are too old.
- The methods compared in the experimental section are quite outdated, with the most recent ones being from 2021. Most of the methods used for comparison are from before 2021.
- How is the proposed method different from other reference-based image rescaling methods?

**Suitability:**

2

---

### Official Review · Reviewer_cZ7G · 2024-05-23

**Rating:** 5
**Confidence:** 3

**Summary:**

this paper propose a novel multi-temporal assisted reference-based image rescaling
framework (RefScale). Specifically, a referencing network is proposed to calculate the similarity map to provide the referencing
condition, which is then injected into the conditional invertible
neural network to guide the information drop at the downscaling
stage and information fusion at the upscaling stage. Additionally,
a low-resolution guidance loss is proposed to further constrain
the data amount of the downscaled LR image. Experiments conducted on both satellite imaging and autonomous driving show
the superior performance of refscale over the state-of-the-art
methods.

**Strengths:**

1. the idea of reference-based scaling is novel
2. this paper has a significant performance improvement compared with existing methods
3. the writings are clear

**Limitations:**

1. as mentioned in the article, the $I^{HR}$ is not available, so the inference and the training steps are different, a clearer explaination or algorithm detail should be provided.

**Suitability:**

3

---

### Official Review · Reviewer_LRjm · 2024-05-24

**Rating:** 2
**Confidence:** 3

**Summary:**

This article introduces a framework called RefScale that aims to solve the problem between low information content and high visual quality in image rescaling. This framework utilizes historical reference images to recover lost information, which is achieved by building a reference network and conditional INN. The authors also designed a similarity loss to guide the training process. The main contributions of the article include introducing multi-period images as reference, proposing the RefScale framework and conducting extensive experimental verification. The results show that it performs well in terms of reconstruction quality and information content.

**Strengths:**

Multi-epoch images are introduced as references in repeated observation scenarios to achieve joint optimization of compression-friendly downscaling and high-quality upscaling.
A novel rescaling framework called RefScale is proposed to indicate the similarity between the reference and the current image through a design generated by reference conditions, and conditionally discard and restore during the rescaling process through a condition-based shrink and enlargement network. mutual information.
Extensive experimental results show that the method exhibits good reconstruction quality and low information content in remote sensing and driving scenarios.

**Limitations:**

The method used in the comparative experiments in this article is not the latest existing method, and it is not an excellent method.
The ablation experiments in this article also do not well prove the indispensability of the method.
The method of the article also does not elaborate on the innovation of the method.
For equation 3 and 5, please explain in detail what they do.

**Suitability:**

2

---

### Meta-Review · Area_Chair_XiDJ · 2024-07-08

**Recommendation:** Accept (Poster)
**Confidence:** 3

**Metareview:**

The paper received mixed reviews but the author's rebuttal, specifically reporting additional results, helped to improve the rating of one reviewer. Since other reviews are still mixed, decision is mainly based on one review which changed from borderline accept to weak accept.